# Phosphodiesterases and Compartmentation of cAMP and cGMP Signaling in Regulation of Cardiac Contractility in Normal and Failing Hearts

**DOI:** 10.3390/ijms23042145

**Published:** 2022-02-15

**Authors:** Gaia Calamera, Lise Román Moltzau, Finn Olav Levy, Kjetil Wessel Andressen

**Affiliations:** Department of Pharmacology, Institute of Clinical Medicine, Oslo University Hospital, University of Oslo, P.O. Box 1057 Blindern, 0316 Oslo, Norway; gaia.calamera@medisin.uio.no (G.C.); l.r.moltzau@medisin.uio.no (L.R.M.); f.o.levy@medisin.uio.no (F.O.L.)

**Keywords:** cardiomyocyte, inotropic response, lusitropic response, beta-adrenergic receptor, GC-A, GC-B, ANP, BNP, CNP, 5-HT4

## Abstract

Cardiac contractility is regulated by several neural, hormonal, paracrine, and autocrine factors. Amongst these, signaling through β-adrenergic and serotonin receptors generates the second messenger cyclic AMP (cAMP), whereas activation of natriuretic peptide receptors and soluble guanylyl cyclases generates cyclic GMP (cGMP). Both cyclic nucleotides regulate cardiac contractility through several mechanisms. Phosphodiesterases (PDEs) are enzymes that degrade cAMP and cGMP and therefore determine the dynamics of their downstream effects. In addition, the intracellular localization of the different PDEs may contribute to regulation of compartmented signaling of cAMP and cGMP. In this review, we will focus on the role of PDEs in regulating contractility and evaluate changes in heart failure.

## 1. Introduction

Phosphodiesterases (PDEs) are the enzymes that degrade both cyclic adenosine 3′,5′-monophosphate (cAMP) and cyclic guanosine 3′,5′-monophosphate (cGMP) to nucleoside phosphates (5′AMP and 5′GMP, respectively). PDEs are categorized in 11 families with subfamilies and splice variants [1]. Based on the different affinities and activities towards cAMP and cGMP, they can be classified as cAMP-selective (PDE4, 7 and 8), cGMP-selective (PDE5, 6 and 9), and dual substrate (PDE1, 2, 3, 10, and 11) PDEs. PDE1–5 and 8–10 are expressed in the heart [2,3]. All PDEs contain a conserved catalytic domain and a variable N-terminal domain. Cyclic nucleotides or PDE inhibitors bind the catalytic site, and differences in single amino acids at this binding site allow development of selective inhibitors for PDE isoforms. The N-terminal region, instead, determines regulation and localization of the enzyme [2].

In this review, we focus on the role of PDEs in regulating cardiac contractility (Figure 1). After covering cardiac contractility and its modulation by cAMP and cGMP, we will discuss in turn the role of the PDEs that are selective for cAMP, the cGMP-selective PDEs and finally the PDEs that can hydrolyze both cAMP and cGMP. Lastly, we will examine some important aspects related to PDEs and contractility, such as changes in PDE function in heart failure and their role in orchestration of cAMP/cGMP signaling in nanodomains.

## 2. Cardiac Contractility

Cardiomyocytes are the key players in cardiac contraction and relaxation. They contain contractile structures composed of the myofilaments, myosin and actin, organized in sarcomeric units. Along the myofilaments, actin monomers are localized with tropomyosin and the complex of troponin T, troponin I, and troponin C (TnT, TnI, and TnC, respectively) [4]. Electric excitation of the sarcolemma initiates cardiac contraction through the process of excitation–contraction coupling, mediated by changes in cytosolic calcium levels [5]. Depolarization of cardiomyocytes triggers opening of L-type Ca^2+^ channels (LTCC) and influx of extracellular Ca^2+^, which in turn triggers the opening of ryanodine receptors (RyR) in the sarcoplasmic reticulum (SR) that release Ca^2+^ into the cytosol. Intracellular Ca^2+^ binds to TnC, leading to a conformational change and binding to TnI; this in turn moves its inhibitory domain away from actin and promotes the displacement of tropomyosin, leaving the actin surface free to bind the myosin heads. Together, these conformational changes lead to formation of cross-bridges along the myosin and actin myofilaments. When the cross-bridges are formed, ATP binds to myosin. Its hydrolysis brings the myosin heads to produce the power stroke, and the myofilaments will slide past each other. This triggers muscle shortening and ultimately contraction of the heart, called systole. Following contraction, cytosolic Ca^2+^ is efficiently removed and the cytosolic Ca^2+^ levels drop, due primarily to the activity of the sarcoplasmatic reticulum Ca^2+^-ATPase (SERCA) and the sarcolemmal Na^+^/Ca^2+^ exchanger, but also the sarcolemmal ATPase and the mitochondrial Ca^2+^ uniporter. As calcium levels drop, calcium bound to TnC is released, and TnI and tropomyosin are returned to a closed state that prevents actin from interacting with myosin. Thus, relaxation of the heart (diastole) can occur [5]. The efficiency of cardiac excitation–contraction coupling and Ca^2+^ removal as well as the contractile proteins themselves are regulated by, e.g., cAMP and cGMP, mainly by phosphorylation of proteins involved in Ca^2+^ handling, such as LTCC, RyR, and the SERCA-inhibiting protein phospholamban (PLB), and sarcomeric proteins, such as TnI, myosin binding protein C (MyBP-C), and titin. Effects that enhance contractility, i.e., the ability of the heart to contract, are called (positive) inotropic effects, whereas effects that enhance cardiac relaxation are called (positive) lusitropic effects.

## 3. Cyclic Nucleotide Signaling and Contractility

Cyclic AMP is a second messenger synthetized by adenylyl cyclase (AC) that can be activated in response to stimulation of G-protein-coupled receptors (GPCRs) coupled to the AC-stimulating G protein G_s_. G_i_-coupled GPCRs, on the other hand, activate the inhibitory G protein G_i_ that inhibits AC activity. Cyclic AMP activates several effector proteins, such as protein kinase A (PKA), exchange proteins directly activated by cAMP (Epac1 and Epac2), cyclic nucleotide-gated ion channels (CNGCs), and Popeye-domain-containing proteins, which regulate a vast array of downstream processes. PKA, as the most important cAMP-activated effector in regulation of contractility, is spatially localized and regulated by a family of scaffolding proteins termed A Kinase Anchoring Proteins (AKAPs). AKAPs anchor PKA to its target and the complexes often include PDEs responsible for regulating local cAMP levels, and phosphatases to tightly regulate PKA-mediated phosphorylation [6,7,8,9,10,11,12].

Cyclic GMP is synthesized by soluble guanylyl cyclase (sGC), activated by nitric oxide (NO), which is in turn synthesized by nitric oxide synthases (NOS); particulate guanylyl cyclases (GCs) can also synthesize cGMP when activated by the natriuretic peptides (NP) atrial natriuretic peptide (ANP), brain natriuretic peptide (BNP), and C-type natriuretic peptide (CNP) [13]. Increase in cGMP by NO/sGC activation can also occur via stimulation of β_3_-AR [14]. In order to mediate its downstream effects, cGMP activates CNGCs and protein kinase G (PKG), which can phosphorylate several targets [15,16].

Cyclic AMP and cGMP activation of PKA and PKG, respectively, modulates contractility through phosphorylation of several proteins. PKA phosphorylation of LTCC and RyR increases their open probability and therefore increases cytosolic calcium levels. Together, this generates a positive inotropic response (PIR). Cyclic AMP also mediates a lusitropic response (LR) through PKA phosphorylation of PLB and TnI. Phosphorylation of PLB removes its inhibitory tone on the SERCA pump, which results in increased pumping of Ca^2+^ into the SR. Phosphorylation of TnI, on the other hand, reduces the sensitivity of the myofilaments for binding Ca^2+^, thus allowing faster Ca^2+^ dissociation. Together, this results in a faster relaxation (lusitropic response) [5]. In situations where cytosolic Ca^2+^ is not limiting, as under cAMP stimulation, PLB phosphorylation will also contribute to a positive inotropic effect through increasing SR Ca^2+^ content. PKG can also phosphorylate PLB and TnI, causing a LR similar to PKA phosphorylation [17,18,19,20,21,22,23,24]. However, contrarily to PKA, PKG phosphorylation of LTCC seems to decrease calcium influx. Thus, due to decreased Ca^2+^ availability due to phosphorylation of PLB and LTCC, combined with TnI phosphorylation, PKG activation is mostly associated with a negative inotropic response (NIR) [25,26,27,28,29]. PKG can also phosphorylate RyR, but no alteration of cardiac RyR activity has been reported [30,31]. Thus, under cGMP stimulation, cytosolic Ca^2+^ may be limiting due to lack of LTCC stimulation, and PLB phosphorylation will mainly contribute to a NIR [23].

## 4. cAMP-Selective PDEs and Compartmentation of cAMP Signaling Regulating Contractility

Intracellular signaling mediated by cyclic nucleotides (CNs) is organized in spatial-temporal nanodomains defined by localization and activities of receptors, adenylyl or guanylyl cyclases, protein kinases and PDEs, and this is defined as compartmentation. The first evidence for compartmentation of second messenger signaling in the heart was published in the 1970s and early 1980s for cAMP, where it was observed that isoproterenol (Iso) stimulation of β-ARs increased contractility, while stimulation with the prostaglandin PgE_1_ did not, despite both increasing cAMP [32,33,34]. It was shown that this difference was related to the intracellular location where cAMP was increased; Iso-stimulated cAMP pools were found in both the soluble and particulate fraction of cardiomyocyte homogenates, whereas cAMP from PgE_1_ was found only in the soluble fraction [35,36].

The GPCR β_1_-adrenergic receptor (β_1_-AR) is the main β-AR in cardiomyocytes. Stimulation of these receptors activates G_s_ that activates ACs, of which AC5 and AC6 are the main isoforms expressed in the heart [37]. Increased cAMP activates PKA that induces a PIR and a LR [38]. β_2_-ARs represent a smaller portion of ARs in the heart and also activate G_s_ (as β_1_-AR). Whereas β_2_-AR stimulation can trigger a PIR, its ability to trigger a LR was disputed [39], although later studies have reported a small LR [40]. The reduced LR might be due to the reported ability of β_2_-ARs to activate both G_s_ and G_i_ but may also be related to the receptor localization. In this regard, β_2_-ARs are found in the T-tubules, and thereby specifically regulate LTCC current [41], while β_1_-AR stimulation produces cAMP both in the T-tubules and at the cell crest [42,43,44,45,46,47,48]. Therefore, β_2_-AR signaling seems very restricted, while the β_1_-AR-stimulated cAMP can diffuse more throughout the cell and reach multiple targets. Similar to β-AR signaling, the serotonin 5-HT_4_ receptors can also give an increase in cAMP and a PIR, at least in failing hearts [49,50,51,52,53].

Cyclic AMP compartments relevant for contractility are regulated by several PDEs. Amongst the cAMP-selective PDEs in the heart, PDE4 is the main PDE, whereas little is known about PDE8.

### 4.1. PDE4

PDE4 is a cAMP-selective PDE with a low K_m_ (1.2–10 µM). It comprises a family of 20 isoforms encoded by PDE4A, B, C, and D genes, of which all except PDE4C are found in the heart [1,54]. The N-terminal region is characterized by upstream conserved regions 1 and 2 (UCR1 and UCR2), which vary between the isoforms and confer them different functional roles and specific subcellular localizations [55,56,57,58,59,60]. When comparing PDE4 activity between rodent and human hearts, many similarities were found, including a rather similar activity [61]. However, due to higher non-PDE4 activities in human heart, PDE4 is responsible for only ~10% of the total cAMP-PDE activity in failing human heart vs. ~50% in normal and failing rat heart [40,61,62,63,64]. This difference implies that data on PDE4 from rodents cannot be applied directly to human myocardium. In this regard, a study on ventricular trabeculae from human failing hearts from patients treated with beta-blockers, either metoprolol [65] or carvedilol [66], showed that inhibiting PDE4 did not affect β_1_/β_2_-AR mediated contractility, while inhibition of PDE3 did. However, also in human failing myocardium, whereas PDE4 inhibition alone had no effect, a role of PDE4 inhibition in enhancement of a 5-HT_4_-mediated positive inotropic response to serotonin was revealed in the presence of PDE3 inhibition [52]. On a similar note, a study in human atrial myocardium revealed a role of PDE4 in controlling arrhythmias when both PDE4 and PDE3 were inhibited [63]. This suggests a role of PDE4 that is secondary to PDE3 in regulating contractility in human heart.

In rodents, PDE4 is the main PDE responsible for hydrolysis of cAMP, when measuring cAMP levels, and together with PDE3 regulates β-AR-stimulated contractility [40,61,63,67,68,69,70]. The importance of PDE4 regulating β_1_-AR-mediated PIR and LR, has been reported in both atria and ventricle [40,71,72]. For β_2_-AR-mediated contractility, there are regional differences in the heart, where PDE4 regulates contractility in the base but not the apex of the ventricle. This could be attributed to several factors, including more regular T-tubular organization and higher PDE4 activity at the base of the heart [73]. In one report using a cAMP biosensor targeted to PLB, β-AR-stimulation was sensitive to inhibition of PDE4 [74]. In another study using a different biosensor targeted to SERCA2A, PDE4 did not regulate β_2_-AR-stimulated cAMP near PLB, whereas PDE2 and PDE3 did [75]. Taken together, this suggests that the β_1_-AR is regulated by PDE4 near PLB. One study has also reported localization of PDE4 near PLB in human failing myocardium [61].

PDE4D3 forms a signalosome together with PKA and RyR regulating the SR-Ca^2+^ leak and in heart failure (HF) reduced local PDE4 gives increased phosphorylation of RyR and greater Ca^2+^ efflux from the SR [76]. Moreover, PDE4D seems to regulate the SR influx of Ca^2+^ by being part of a complex with SERCA, PKA, and PLB [77]. PDE4B is involved in the regulation of LTCC, which allows Ca^2+^ entrance in the cell when phosphorylated by PKA [78,79]. Recent work from the Nikolaev lab, using a novel biosensor located near RyR, showed that PDE4 was the main PDE responsible for degrading cAMP near RyR, and that this PDE4 activity is reduced in HF due to a redistribution of PDE4D leading to lower amounts associated with RyR [80]. This could contribute to the altered PDE4 regulation of contractility in HF [40].

In some reports on human, porcine, and rodent cardiac ventricle, β_1_-AR-, β_2_-AR-, or 5-HT_4_-mediated contractility was not enhanced by inhibiting PDE4 alone, despite increased levels of cAMP. Rather, simultaneous inhibition of both PDE3 and PDE4 was needed to obtain an increased contractility, both in normal and failing hearts [40,52,69,71,72,81,82,83,84]. Also, upon 5-HT_4_ receptor stimulation, inhibition of both PDE3 and PDE4 significantly increased cAMP levels in porcine left atrium [85] and in human atrial cardiomyocytes, where the effect of PDE3 and PDE4 inhibition seemed more pronounced and thus was able to restore a blunted 5-HT response in cardiomyocytes from patients with persistent atrial fibrillation [86]. Together, this suggests that both PDE3 and PDE4 are responsible for regulating contractility and behave in a redundant manner. In support of PDE redundancy, using in silico modeling, Zhao et al. showed that under Iso-stimulation the most active PDE is PDE4, while PDE2 activity increased to a smaller degree and even smaller for PDE3. However, when PDE2 was inhibited, they showed that its activity dropped together with a rise in PDE4 activity. Similarly, when PDE4 was inhibited, increased PDE2 activity compensated for the loss in cAMP hydrolytic capacity [87].

### 4.2. PDE8

PDE8 is a cAMP-selective phosphodiesterase and has two isoforms: PDE8A and PDE8B [88]. The regulatory domain comprises a REC domain (similar to the “receiver” domains of bacterial two-component signaling systems) [89] and a PAS domain (an acronym for Period, Arnt, and Sim), however, little is known about the function of these regulatory domains [90]. PDE8 is expressed in cardiomyocytes, but its function is not completely understood. One study shows that PDE8A^−/−^ cardiomyocytes have larger Ca^2+^ transients, increased LTCC current, and increased cardiomyocyte contractility upon β-AR activation, but also larger Ca^2+^ spark frequency in the absence of receptor activation [91]. This suggests alteration of excitation–contraction coupling through PDE8 inhibition. Since PDE8 is not sensitive to the widely used “general” PDE inhibitor IBMX, caution should be taken when interpreting studies with a lack of the IBMX effect, as there might still be a hidden effect of PDE8. For example, it was reported that β_1_-AR caused a widespread cAMP signal, whereas β_2_-AR caused a much more localized cAMP signal [45]. If this difference was, at least in part, due to cAMP degradation by PDEs, one would think that PDE inhibition should make the β_2_-AR cAMP signal more similar to that of β_1_-AR. However, the difference was insensitive to PDE inhibition by IBMX, which at the time would indicate that PDEs were not involved. However, an intriguing possibility is that PDE8 could be involved in the difference, and that PDE8 inhibition, maybe in addition to IBMX, would make the β_2_-AR signal more like the β_1_-AR signal.

## 5. cGMP-Selective PDEs and Compartmentation of cGMP Signaling Regulating Contractility

One of the first indications of cGMP compartmentation was seen in the early 1990s when cGMP from NO was found to be in both the soluble and the particulate fraction, while cGMP from the pGC was present only in the particulate fraction [92]. In rat hearts, NO-dependent sGC activation does not modulate IR or LR [93,94,95,96], whereas CNP-stimulation of GC-B does [17,18,19,20,21,22,23,24,29,53,62,64,97,98]. In most studies the natriuretic peptides ANP and BNP activating GC-A show no effect on contractility [17,18,24,29,53,62,98], except for a few studies showing a NIR [20,28,99]. This indicates that even though the particulate GCs (GC-A and GC-B) are both in the plasma membrane, somehow they still increase cGMP in different subcellular compartments. Nevertheless, ANP and BNP seem to mediate PKG phosphorylation of LTCC, thus decreasing the calcium current [25,27,28,29]. The natriuretic peptide CNP can elicit a NIR and a LR through PKG phosphorylation of PLB (leading to increased SERCA2 activity) and TnI, in normal rat and mouse hearts [17,18,19,20,21,22,23,24], but also in rat failing hearts [23,29,64]. In addition, an initial PIR has been reported and linked also to PLB phosphorylation and increase in PKG-mediated calcium transients [18,19,23,100]. However, the mechanism behind this initial PIR is not completely understood. It could be due to phosphorylation of PLB happening as a first step, which would increase cytosolic calcium sequestration in the SR, and therefore promoting the next contraction cycle. This would imply a delay in TnI phosphorylation by PKG, which then gives the NIR. Alternatively, cGMP increase could initially inhibit PDE3 and increase cAMP-mediated contractility (see “Dual-substrate PDEs and compartmentation of cyclic nucleotide signaling regulating contractility”), followed slightly later by PKG phosphorylation of the downstream targets, which could overcome the initial PIR and elicit a NIR. These speculations, however, would require further investigation. CNP is found to reduce LTCC current in failing and normal cardiomyocytes through activation of PKG [26,29]; this might contribute to the NIR, but additional mechanisms must be involved as BNP, also reducing LTCC current, does not induce a NIR [29].

The PDEs that selectively hydrolyze cGMP in the heart are PDE5 and PDE9.

### 5.1. PDE5

PDE5 selectively degrades cGMP and has three isoforms (PDE5A1–3) expressed in human hearts [101]. The regulatory domain contains two GAF (cGMP-binding PDEs, Anabaena adenylyl cyclase, and Escherichia coli Fh1A) domains and a phosphorylation site. Cyclic GMP binds to the GAF-A domain with high affinity, causing an allosteric increase in enzymatic activity. The enzymatic activity and thus cGMP degradation is also increased by phosphorylation of PDE5 by PKG [102]. In adult cardiomyocytes, PDE5 is expressed at low levels and localizes to the Z-disc where it degrades cGMP mainly from the NO/sGC pathway [103,104,105]. In unstimulated cells, using an untargeted FRET biosensor, PDE5 shows low activity towards cGMP [24,106].

Inhibiting PDE5 does not affect basal contractility. However, when activating all three β-ARs, PDE5 inhibition (by sildenafil) reduced contractility in normal mouse heart involving increase in NOS3-NO-sGC-originated cGMP [97,103], as well as in human healthy volunteers [107]. In order to elucidate the signaling behind the negative effect of sildenafil on contractility, Isidori et al. showed that consequent cGMP augmentation led to activation of PDE2 and thus decrease in cAMP levels; this was associated with decreased β_2_-AR-mediated contractility after PDE5 inhibition [108]. In addition, Schobesberger et al. showed that PDE5 regulated a cGMP-pool generated by NOS3-sGC activation subsequent to β_3_-AR activation, at caveolae-enriched membrane fractions in T-tubules. This cGMP pool was then responsible for a decrease in cAMP levels through PDE2 activation [109]. These data fit with previous data where β_3_-ARs were found to elicit a NIR and positive LR via eNOS and cGMP [14,110]. In a chronic HF model, a role of PDE5 in regulating contractility was revealed only after simultaneous PDE3 inhibition, which increased the maximal NIR to CNP, indicating a redundancy between these two PDEs [64].

### 5.2. PDE9

PDE9A is the PDE with the lowest K_m_ for cGMP (0.70–0.17 µM) and it was recently discovered to be expressed in the heart, localized to the SR (but not Z-disc, where PDE5 was located), and regulating cGMP generated after GC-A stimulation, but not from stimulating sGC [2,105]. Moreover, PDE9 is localized in the mitochondria in cardiomyocytes [111]. We are not aware of any literature where PDE9 inhibition regulates cGMP-mediated contractility. However, the availability of novel PDE9 inhibitors may clarify whether PDE9 regulates inotropic and lusitropic responses.

## 6. Dual-Substrate PDEs and Compartmentation of Cyclic Nucleotide Signaling Regulating Contractility

Although the cAMP- and cGMP-pathways modulate contractility differently, as described above, increased concentrations of cGMP can also modulate cAMP levels, by affecting the activity of certain PDEs. Cyclic GMP binds to a GAF domain on PDE2 to increase the enzymatic activity, and thereby reducing cAMP responses, whereas at PDE3 cGMP is hydrolyzed slowly and this inhibits cAMP degradation, increasing cAMP-mediated responses. Thereby, PDEs that can hydrolyze both cAMP and cGMP are responsible for so-called cross-talk between the two signaling pathways. The compartments relevant in contractility that involve such cross-talk are not easy to differentiate from the specific cAMP/cGMP compartments described above. Further investigation with focus on local cyclic nucleotide fluctuations and on how subcellular compartmentation occurs, will help to resolve this.

The dual-substrate PDEs also have important roles towards the two single pathways, and in the next section, we describe how PDE1, PDE2, and PDE3 contribute to modulate contractility by regulating either cAMP or cGMP separately, but also how they modulate contractility by creating cGMP/cAMP cross-talk.

### 6.1. PDE1

The PDE1 family includes the isozymes PDE1A, PDE1B, and PDE1C [112]. These are Ca^2+^/calmodulin (CaM)-dependent enzymes and their regulatory domain contains two CaM binding sites, two phosphorylation sites, and an inhibitory region, which retains the enzyme in its inactive configuration when the calcium levels are low [113,114]. CaM binding to PDE1 elevates its activity, whereas phosphorylation by PKA or CaMKII reduces the affinity for Ca^2+^/CaM and therefore its activation [115,116]. PDE1 enzymes can hydrolyze both cAMP and cGMP. PDE1A and PDE1B catalyze cGMP degradation with higher K_m_ than for cAMP, while PDE1C has equal affinity [112]. Among the three isoforms, PDE1A and PDE1C are found in the heart. However, PDE1C is the predominant isoform in the human heart and specifically localizes to the M and Z lines in cardiomyocytes, while PDE1A is the predominant isoform in mice and rats [117,118,119]. Expression of PDE1 was increased in failing hearts [119,120].

Most studies report data on the functional role of PDE1 in rodent hearts and do not investigate its specific effect on contractility. Recently, the group of David Kass has shown the role of PDE1C in rabbit and dog hearts, which, similar to human hearts, express predominantly this isoform and show positive inotropic and chronotropic effects after inhibiting PDE1. These effects are due to PDE1C activity over a soluble AC-cAMP nanodomain that regulates LTCC current via PKA, but does not involve modulation of SERCA current, phosphorylation of PLB, TnI, and myosin binding protein C, like in the β-AR-mediated contractility [119,121].

### 6.2. PDE2

PDE2 is a dual substrate phosphodiesterase with similar high V_max_ and high K_m_ for both cAMP and cGMP [122,123]. The isoform PDE2A has three splice variants that have either soluble (PDE2A1) or particulate (PDE2A2, PDE2A3) distribution [122,123,124,125,126]. PDE2 is found in human hearts [127] and in cardiomyocytes it localizes in the cytosol, plasma membrane, sarcomeric Z line, mitochondria, and nuclei [110].

The regulatory region contains two GAF domains, A and B [128]. Cyclic GMP binding to the GAF-B domain raises the esterase activity for both cAMP and cGMP [129]. Thus, this PDE is a key factor in the cGMP/cAMP cross-talk signaling and in the regulation of cardiac contractility [130].

#### 6.2.1. PDE2 Regulation of cAMP Relevant for Contractility

In rat and mouse cardiomyocytes PDE2 contributes to a low fraction of the total cellular cAMP-PDE activity [64,110], while in humans with HF, it contributes ~25% of the total cAMP-PDE activity [61]. There is solid evidence that PDE2 has a relevant role in modulating β-AR-stimulated cAMP pools. Accordingly, PDE2 inhibition increases LTCC currents and contractility, both in the absence and presence of β-AR-stimulation [67,131,132]. However, in other studies PDE2 inhibition did not influence basal or β-AR-elevated global cAMP, LTCC currents, or inotropic response [40,133]. In HF, PDE2 inhibition increased LTCC currents and sarcomere shortening in the absence and presence of β-AR stimulation [134]. In mice expressing either an untargeted or PLB-targeted cAMP sensor, PDE2 limits β-AR-stimulated cAMP both in the whole cell and in the compartment surrounding PLB and the activity was increased around PLB in a mild hypertrophic model [74].

#### 6.2.2. PDE2 Regulation of cGMP Relevant for Contractility

Total cGMP levels seem to be mainly controlled by PDE2 both in normal [106] and failing hearts [64]. However, when using an untargeted FRET biosensor in cardiomyocytes stimulated with CNP, PDE2 inhibition gave a smaller increase in cGMP compared to PDE3 inhibition [24]. Instead, PDE2 restricts ANP-stimulated cGMP which is localized at the T-tubules preventing far-reaching signals [104,135]. Accordingly, it seems that PDE2 activity is relevant in restricted compartments, in line with the observation that PDE2 inhibition increased CNP-stimulated cGMP levels measured by targeted biosensors near PLB and TnI [24]. Regulation of TnI and PLB is linked to modulation of the LR and in normal heart PDE2 also regulates CNP-induced LR, similarly to PDE3, since inhibition of either PDE resulted in a sensitized response [24]. In a rat HF model, PDE2 inhibition desensitized the CNP-induced NIR but had no effect on the lusitropic response [64].

#### 6.2.3. PDE2 and cGMP/cAMP Cross-Talk Relevant for Contractility

Considering that PDE2 is a dual substrate PDE, and that cGMP can bind to the PDE2-GAF domain and increase its enzymatic activity, the role of PDE2 in cGMP/cAMP cross-talk has been investigated. Specifically, cGMP-increase after sGC stimulation reduces LTCC current through activation of PDE2 and this is mediated by a decrease in cAMP and PKA phosphorylation of LTCC [133,136,137,138]. Activation of PDE2 by cGMP from sGC is also responsible for decreased contractility after β_1_- and β_2_-AR activation [133], and such effect seems to derive from local cGMP increases in nanodomains defined by PKA RII-cAMP biosensors [139]. Mongillo et al. investigated cGMP/cAMP cross-talk by examining the role of all three β-ARs in adult cardiomyocytes. They showed that such stimulation involves a mechanism where activated β_3_-ARs trigger NO release and consequent cGMP augmentation and that these pools of cGMP are responsible for PDE2 activation and thereafter reduction in β_1_- and/or β_2_-AR-stimulated cAMP levels and PIR [110]. Supporting this mechanism, Schobesberger et al. further confirmed the localization of β_3_-ARs in the caveolae and the related cGMP pools in the T-tubules, which are locally regulated mostly by PDE2 and PDE5 [109], and stimulation by NO-donor after β-AR-stimulation further decreased global cAMP levels in HF [134].

CNP can also, through cGMP-mediated activation of PDE2, decrease cAMP levels and thereby decrease β-AR-stimulated LTCC current [138,140]. Even though GC-A-stimulated cGMP microdomains seem to act differently than those stimulated by GC-B and do not seem to potentiate β-AR-stimulation of the cAMP pathway in several studies [53,62,98,106], some studies reported that GC-A stimulation can either reduce β-AR-mediated cAMP via PDE2 activation in nanodomains defined by a PKA RII-targeted biosensor [139] or increase the β-AR-mediated contractile effects in a mild hypertrophic model after TAC (transverse aortic constriction) [141]. Real-time measurements of cAMP levels with a caveolin-targeted biosensor indicated a reorganization of PDE2 and PDE3 between β_1_- and β_2_-AR compartments, revealing an ANP-induced increase of the positive inotropic effect of β-adrenergic stimulation in a mild hypertrophic model through PDE3 inhibition [141]. In this model, PDE2 activity was reduced around the β_1_-AR microdomain and increased around the β_2_-AR microdomain, therefore, ANP-stimulated cGMP will switch from reducing cAMP levels via PDE2 activation (in physiological conditions) to increase cAMP levels via PDE3 inhibition (in a disease model) relatively to the β_1_-AR microdomain [141].

### 6.3. PDE3

PDE3 has two isoforms, PDE3A and PDE3B, both expressed in the heart [142,143,144]. PDE3 hydrolyzes both cAMP and cGMP with similar high affinities (K_m_ cAMP 0.16 µM; K_m_ cGMP 0.09 µM), while the V_max_ is >10-fold higher for cAMP. Hence, when cGMP binds, it is hydrolyzed slowly and this inhibits cAMP degradation [1,145]. The N-terminal structure contains hydrophobic loops that allow for membrane insertion of the enzyme, and PDE3B is localized to T-tubule membranes while PDE3A is mainly found in SR membranes [146]. The latter forms a complex with SERCA, PLB, and AKAP18δ, and when PKA phosphorylates PDE3, cAMP hydrolysis is increased therein [147,148,149]. Consequently, PDE3A1 is regulating the cAMP-mediated regulation of Ca^2+^ re-uptake in the SR [67,149,150,151].

#### 6.3.1. PDE3 Regulation of cAMP Relevant for Contractility

PDE3 is responsible for hydrolyzing global cellular cAMP levels in hearts from normal and HF animals, to a lower extent than PDE4, which is the main cAMP-PDE in rodents [40,68,74]. As mentioned previously, in humans PDE4 has a lower contribution to the total PDEs activity, and therefore PDE3 is the most important PDE degrading cAMP in human hearts [61,63,65]. Accordingly, PDE3 was found in the SR [152] and shown to be the main PDE regulating contractility in human atrium [153] and ventricle [52,62,65,154]. In rats, PDE3 is found to regulate both β-AR- and 5-HT_4_ receptor-mediated PIR and LR [40,52,98] and also regulates calcium transients in the absence and presence of β-AR stimulation and its inhibition causes an increase in sarcomere shortening [131,155].

Using targeted FRET-based biosensors, PDE3 subcellular activity was found to prevail around cAMP nanodomains defined by a PKA RI-targeted biosensor [139] and regulate β_2_-AR signaling in caveolin-enriched compartments such as T-tubules [141]. However, in a mild hypertrophic model, PDE3 hydrolyzing activity towards β_2_-AR-cAMP at the plasma membrane was reduced, with no changes in global expression, but with clear reduction in the caveolin-rich fractions [141]. Similarly, the role of PDE3 in the regulation of β-AR-mediated cAMP and LTCC current might prevail in hypertrophic compared to normal cardiomyocytes, due to reduced PDE4 activity [84]. Therefore, due to the apparent redundancy between PDE3 and PDE4, inhibiting both PDEs triggers a greater effect on basal, β-AR- and/or 5-HT_4_-receptor-induced cAMP levels, calcium, PIR and LR, compared to inhibiting either PDE alone [40,52,69,84,156].

#### 6.3.2. PDE3 Regulation of cGMP Relevant for Contractility

With regard to cGMP, PDE3 regulates global cellular cGMP levels and cytosolic cGMP measured by a non-targeted biosensor [106] and contractility, both in normal and failing hearts [24,64]. Using targeted FRET-based biosensors PDE3 was shown to constrain CNP-stimulated cGMP at the plasma membrane, PLB, and myofilament compartments [24,135]. Inhibition of PDE3 increased the CNP-mediated LR in normal heart muscle, and the CNP-mediated NIR and LR in HF models [24,64].

#### 6.3.3. PDE3 and cGMP/cAMP Cross-Talk Relevant for Contractility

The natriuretic peptide CNP can modulate contractility by affecting cAMP signaling through cGMP-mediated inhibition of PDE3, causing a PIR mediated by β_1_-AR and β_2_-AR in normal and failing rat left ventricle [53,62,98]. 5-HT_4_ receptor-induced PIR was also shown to be increased by CNP-induced PDE3 inhibition in both failing rat left ventricle and in porcine left atrium [53,81]. In HF, constitutive NO/sGC activation increased 5-HT_4_-mediated PIR (through PDE3 inhibition), but also reduced β_1_-mediated IR [53]. The reduction in β_1_-mediated cAMP and IR by NO/sGC has been shown by others as discussed above under “PDE2 and cGMP/cAMP cross-talk relevant for contractility” [103,109,110,133,134,139]. The opposing effects of sGC stimulation on β_1_-AR and 5-HT_4_ receptors may indicate different compartmentation of PDEs in their signaling pathways. As also addressed above, ANP potentiated the β-AR-stimulated PIR, presumably through PDE3 inhibition in a mild hypertrophic model, but not in Sham [141]. However, in other studies GC-A stimulation did not affect β_1_-AR-dependent contractility, neither in a chronic HF model nor in sham-operated animals [53,98].

### 6.4. PDE10

PDE10 is a dual PDE discovered in 1999 [157,158,159] and its role in the heart was only recently explored by Chen et al., where they showed upregulation of PDE10 in failing hearts and beneficial effects of PDE10 inhibition against hypertrophy and fibrosis. They also showed a restored contractile function in a mouse model of congestive HF, where the PDE10 gene is knocked down [3]. Therefore, it is not yet clear whether it directly regulates cardiac contractility.

## 7. Modulation of PDEs in Heart Failure

In HF the heart fails to provide adequate blood flow/pressure to meet the body’s demands and this is associated with alterations in the cardiac structure, function, rhythm, and conduction. In HF, the body responds by activating compensatory mechanisms, such as salt and water retention, increase in noradrenaline and adrenaline (sympathetic activation), and increase in several hormones, such as angiotensin II, aldosterone, endothelin, and natriuretic peptides. Some of these changes will in the short term improve cardiac contractility and cardiac output, but in the long term some of these will worsen the prognosis [160,161]. Of these compensatory mechanisms, there are specific signaling pathways that regulate the beneficial and detrimental effects, and it is crucial to understand such signaling to improve HF treatment. Currently, HF is routinely treated with β-blockers, angiotensin converting enzyme (ACE) inhibitors/angiotensin receptor blockers (ARBs), mineralocorticoid receptor antagonists, sodium–glucose co-transporter 2 inhibitors and sometimes angiotensin receptor–neprilysin inhibitor, as well as diuretics when needed [162].

Despite the fact that cAMP and cGMP have many similarities in the signaling pathway and have similar protein targets, they can mediate different and sometimes opposite effects on contractility in the heart, as described above. In HF there is increased adrenergic activity stimulating β-ARs and increased natriuretic peptides. Long-term stimulation of β-AR is detrimental in chronic heart failure and β-blockers are a cornerstone in HF management [162,163]. Increasing cGMP, on the other hand, seems to be beneficial in the cardiovascular system and can mediate antifibrotic and antihypertrophic effects, vasodilation, natriuresis, and modulate cardiac contractility [164,165,166,167,168,169]. Stimulating sGC was protective against HF in mice [170], and in humans, stimulating sGC is beneficial in treatment of pulmonary arterial hypertension and HF with reduced Ejection Fraction (HFrEF) [171,172]. Inhibiting degradation of natriuretic peptides by the neprilysin inhibitor sacubitril showed improvements in HFrEF when administrated in combination with the angiotensin receptor blocker valsartan [173]. This drug combination was also shown to possibly improve HF with preserved Ejection Fraction (HFpEF) in certain subgroups [174]. In HFpEF, increased cardiac stiffness could impair diastolic filling. PKG phosphorylation of the structural sarcomeric protein titin decreases cardiac stiffness [175] and GC-B-stimulated cGMP increased titin phosphorylation and decreased cardiomyocyte stiffness in rat cardiomyocytes and during early phases of pressure overload in mice [169,176]. PKA also phosphorylates titin, reducing human cardiomyocyte stiffness, suggesting that increasing cAMP/PKA activation could have beneficial effects in HFpEF [177,178,179]. Taking these considerations together, modulation of PDEs that reduce cAMP levels but increase cGMP, could be a good strategy for HF treatment. However, in light of the different compartments regulating cardiac function, the reality is more complicated. In the following section, we discuss how PDEs change in HF and whether overexpression or inhibition of PDEs could represent possible pharmacological treatments (Figure 2; Table 1).

### 7.1. PDE2

PDE2 expression increases in HF [134,180]. However, it is not clear whether activation or inhibition would be beneficial. During HF, the β-adrenergic pathway is highly activated and can trigger detrimental effects in the heart, hence activating PDE2 would counteract such effects. Accordingly, PDE2 overexpression reduces β-AR-stimulated LTCC current and prevents the cAMP-mediated inotropic and lusitropic response and hypertrophy [134]. In addition, increase in PDE2 expression prevents arrhythmias and is beneficial after myocardial infarction in mouse models, maintaining cardiac contractility [192]. On the contrary, overexpression of PDE2 showed increased stiffness in connective tissue [193] and this might be due to diminished anti-fibrotic effects of cGMP and cAMP [194,195] or to decreased phosphorylation of titin through PKA or PKG [196]. Inhibition of PDE2 enhancing cAMP/cGMP might be beneficial against cardiac hypertrophy [170,197,198] and might counteract development of HF.

### 7.2. PDE3

In heart failure, alteration in PDE3 expression is not that clear. Some studies show a reduction in both humans and animal models [181,182,183,184], whereas other studies in animal models show an increase [185] or no change [98]. These discrepancies could be due to different HF models and different species. Inhibition of PDE3 was previously considered as a potential treatment for heart failure, as it would increase cAMP and ameliorate cardiac contractility and relaxation. Therefore, several PDE3 inhibitors were studied for chronic heart failure treatment (milrinone, amrinone, enoximone, and cilostazol). However, the initial promising results failed in clinical studies showing increased mortality [199,200]. Years earlier, a downregulation of β-ARs was observed in failing hearts, together with higher levels of inhibitory G proteins [201]. It was then postulated that chronic sympathetic stimulation of the heart triggers compensatory mechanisms aiming to reduce energy-consuming mechanisms, resulting in cardiac failure. This explained why treatment with PDE3 inhibitors was deleterious in chronic HF and a break-through in the HF therapy occurred with the successful outcome of using β-blockers [154,163]. Currently, milrinone is used for acute decompensated heart failure [162,202] and it is found to be well tolerated in patients with HFpEF, but further studies in these patients are required [203]. Since mouse studies knocking down PDE3 show different effects of PDE3A and PDE3B in the heart, this suggests that PDE3A is the more important isoform in regulating contractility [146,204].

### 7.3. PDE4

In HF expression of both PDE4A and PDE4B, as well as PDE4 activity, is reduced and accompanied by heart dysfunction [40,63,76,79,184,186]. In support of decreased PDE4 levels, inhibition of PDE4 does not affect β-AR-mediated contractility in human failing hearts [52,63,65,66,153,205], nor in failing rat ventricles [40], but simultaneously inhibiting PDE3 and PDE4 showed an effect greater than PDE3 inhibition alone in failing human and rat ventricle [40,52] as well as human atrium [63].

In a recent study, PDE4B overexpression both decreased contractility and lusitropic responses through reduced phosphorylation of PLB, TnI, and MyBPC [206]. In addition, using several HFrEF models, PDE4B overexpression (either through mild transgenic overexpression or heart-specific AAV9-induced expression) was shown to be cardioprotective [206]. It would therefore be interesting to determine whether cardiac PDE4B overexpression in humans is beneficial in HFrEF.

### 7.4. PDE5

PDE5 activity is enhanced in HF, both in humans and in animal models [166,187,207,208,209] and its localization is rather diffuse [187,188,189,190,210], with a possible functional retargeting to the ANP-stimulated cGMP pool [191]. Long-term treatment with the PDE5 inhibitor tadalafil restored the cardiac response to β-AR stimulation, improved contractility, and reversed the T-tubule loss typically associated with HF remodeling [211,212]. In humans, long-term PDE5 inhibition has shown beneficial effects in HFrEF but has failed to show clinical benefits in HFpEF [213,214,215,216,217]. However, currently it is still discussed whether PDE5 is expressed in cardiomyocytes, and therefore its function in the myocardium is controversial, as reviewed recently by Dunkerly-Eyring and Kass [218].

### 7.5. PDE9

The expression of PDE9 in the heart is low, but it is upregulated in human left ventricular biopsies from HFpEF and aortic stenosis patients and in animals with HFrEF [105]. Inhibiting or knocking out PDE9 improves ejection fraction and left ventricular diastolic filling in animal models of HF and is protective against hypertrophy and fibrosis [105,219,220].

## 8. Orchestration of Cyclic Nucleotide Signaling in Nanodomains

For many years, studies on the intracellular signaling modulating contractility have focused on measuring global cAMP levels in relation to a specific functional response. However, we now understand more of the organization of the compartmented signaling networks relaying responses, from GPCR activation to cAMP signaling in subcellular nanodomains (recently reviewed by Zaccolo et al. [221]). Consequently, instead of measuring global cyclic nucleotide levels, investigating local changes in cAMP and cGMP is needed in order to understand the mechanisms behind specific downstream effects. This has largely been achieved by the development of FRET-based biosensors, which can measure cAMP and cGMP in real time in living cells, and through targeting these to specific subcellular compartments. Compartmentation can also explain the disparities between which PDE was important for the regulation of global cellular cAMP and cGMP levels, versus which PDE was involved in regulation of cAMP- and cGMP-mediated functional responses. For example, PDE3 is the most sensitive for both cAMP and cGMP. Considering that basal cAMP concentrations are around 1 µM and cGMP concentrations around 10 nM [74,106,222], PDE3 in unstimulated conditions would mainly hydrolyze cAMP. Together with PDE3, also PDE4 has been shown to constrain basal cAMP levels. However, the affinity of PDE4 for cAMP is more than 10-fold lower than PDE3 and also above the basal cAMP levels. Moreover, PDE2 has even lower affinity for cAMP and should therefore not regulate basal levels, unless its total capacity is very high and thus relevant also well below the K_m_. However, inhibition of PDE2 increased cell shortening at basal conditions [134]. Similar disparities are seen for cGMP-hydrolyzing PDEs. After CNP stimulation, PDE2 inhibition greatly increased global cGMP levels, significantly more than PDE3 or PDE5 inhibition [64], but when using an untargeted cGMP sensor, PDE3 inhibition increased CNP-stimulated cGMP levels to a greater extent than PDE2 inhibition [24]. Targeting this biosensor to TnI or AKAP18δ demonstrated that PDE2 and PDE3 similarly regulate cGMP at these nanodomains and similarly regulate CNP-induced LR [24]. These observations would suggest that cGMP is locally concentrated around the target with specific PDEs regulating the local cGMP levels and inhibiting these PDEs would disrupt the compartmentation, causing a global cGMP increase.

Localization of signaling in subcellular compartments could also explain the discrepancy between global cAMP concentrations at basal conditions, which are around 1 µM, and PKA affinity for cAMP, which is ~100 nM [222,223,224,225]. In subcellular compartments, the local cAMP concentrations would be restricted by the activity of PDEs. However, cAMP is known to diffuse at a higher rate than the catalytic rate of PDEs [226,227]. Therefore, it is still puzzling how compartments can occur. Additional understanding on how cyclic nucleotides and PDEs are organized in nanodomains, has been provided by in silico modeling approaches. A recent review on modeling approaches for cAMP compartmentation nicely describes the advances that such studies have provided in better understanding the spatial and temporal organization of the cyclic nucleotide systems in cardiomyocytes. Using spatial and stochastic modeling, it became clear that cAMP could not be freely diffusing in the cells [228]. Recently, it was demonstrated that cAMP diffusion is restricted in cells and in addition that PDEs create nanodomains of low cAMP [229]. The limited diffusion was due to immobile cAMP binding sites. Another study demonstrated that cAMP is buffered by biomolecular condensates made up by PKA RI subunits that form liquid–liquid phase separation (LLPS) in the cytosol. These LLPSs are able to maintain the free cAMP levels low, so that few cAMP molecules can reach the local targets [230]. Thus, LLPS buffering and PDEs create nanodomains with low cAMP that can protect PKA from unwanted activation, until receptor stimulation enhances cAMP further to overcome both buffering and PDE capacity and thus activate PKA [229,230].

## 9. Summary

Since PDE inhibitors were first used for treatment of HF, many advances have been made in understanding the role of PDEs in cardiac contractility. However, there are some ongoing challenges to consider. For example, a large number of studies are conducted using animal models where PDE activity and expression can be different from those in humans, and thus caution is needed when interpreting the data. As we appreciate more the importance of signaling compartmentation, localization of specific PDEs is also of relevance in modulation of contractility.

Real-time measurements of cAMP and cGMP using targeted biosensors has been pivotal in understanding how PDEs contribute to mediate contractility at the subcellular level and helped to uncover signaling compartmentation. Yet, there are still some questions and challenges remaining: how do PDEs efficiently degrade cAMP/cGMP to protect protein kinase from unwanted cellular activation? How does stimulation of a specific receptor trigger the activation of certain downstream targets and in some compartments rather than others? Moreover, how are the compartments involving cGMP/cAMP cross-talk organized vs. the single cAMP/cGMP compartments?

Using targeted biosensors combined with in silico modeling will help to understand the compartments in a cell, which modulators are involved, the organization of cAMP/cGMP local nanodomains and their size, and how these are affected in cardiac dysfunction.

Heart failure has been associated with changes in either expression or localization of PDEs. In addition, there is evidence that some PDEs can behave in a redundant manner. How one PDE may compensate for changes in the activity of another PDE during signaling modulation or whether this redundancy changes during HF, remains to be fully elucidated.

## Figures and Tables

**Figure 1 ijms-23-02145-f001:**
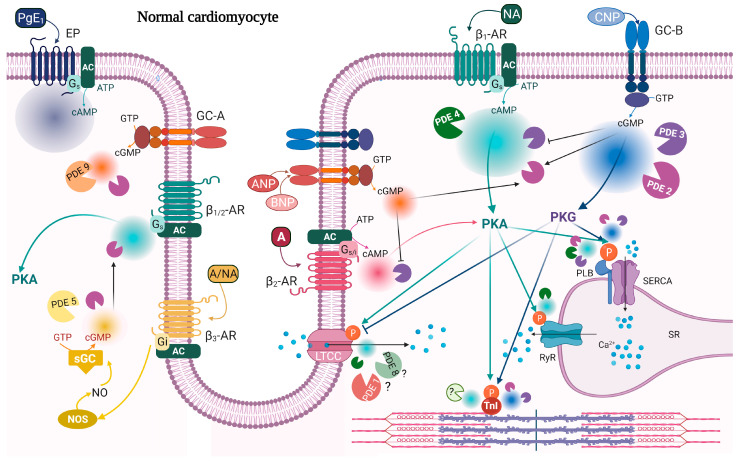
cAMP/cGMP signaling and PDEs modulating contractility in normal cardiomyocytes. β_1_-AR stimulation causes a widespread rise in cAMP (cyan), which activates PKA phosphorylation of PLB, TnI, RyR, and LTCC. These events together lead to a positive inotropic response and a positive lusitropic response. Global and localized cAMP pools are under the control of PDE2 (purple), PDE3 (violet), and PDE4 (dark green) activity. β_2_-AR stimulation increases cAMP (pink), degraded by PDE2, and which affects contractility, but to a lower extent compared to β_1_-AR stimulation. CNP stimulation of GC-B gives a widespread increase in cGMP (blue) and PKG activation; PKG phosphorylates PLB and TnI and decreases LTCC current. These events contribute to a negative inotropic response and a positive lusitropic response. Global and local cGMP are degraded by PDE2 and PDE3. ANP stimulation of GC-A triggers a more restricted cGMP pool (orange) which does not seem to have a major direct role in contractility, and it is under the control of PDE2 and PDE9. cAMP/cGMP cross-talk is illustrated with black arrows indicating cGMP activation of PDE2 or inhibition of PDE3 activity towards cAMP. β_3_-AR stimulation can indirectly modulate contractility by increasing an NO-dependent cGMP pool which activates PDE2 and decreases β_1/2_-AR-stimulated contractility. This pool of cGMP (yellow) is degraded by PDE2 and PDE5. Prostanoid receptors (EP) stimulated by prostaglandin E_1_ (PgE_1_) increase cAMP with no effect on contractility. Shown also are receptors demonstrated to be present in T-tubule. Question marks indicate either unknown PDE or that the function of that PDE is not clear. A, adrenaline; AC, adenylyl cyclase; ANP, atrial natriuretic peptide; ATP/GTP, adenosine/guanosine triphosphate; BNP, brain natriuretic peptide; cAMP/cGMP, cyclic adenosine/guanosine 3′,5′-monophosphate; CNP, C-type natriuretic peptide; EP, prostanoid receptor; GC-A/B, guanylyl cyclase A/B; G_s_/_i_, stimulatory/inhibitory G protein; LTCC, L-type calcium channel; NA, noradrenaline; NO, nitric oxide; NOS, nitric oxide synthases; p, indicates phosphorylation; PDE, phosphodiesterase; PgE_1_, prostaglandin E_1_; PKA/PKG, protein kinase A/G; RyR, ryanodine receptor; SERCA, sarcoendoplasmic reticulum (SR) calcium ATPase; sGC, soluble guanylyl cyclase; TnI, troponin I; β-AR, beta-adrenergic receptor. Created with BioRender.com, accessed on 7 February 2022.

**Figure 2 ijms-23-02145-f002:**
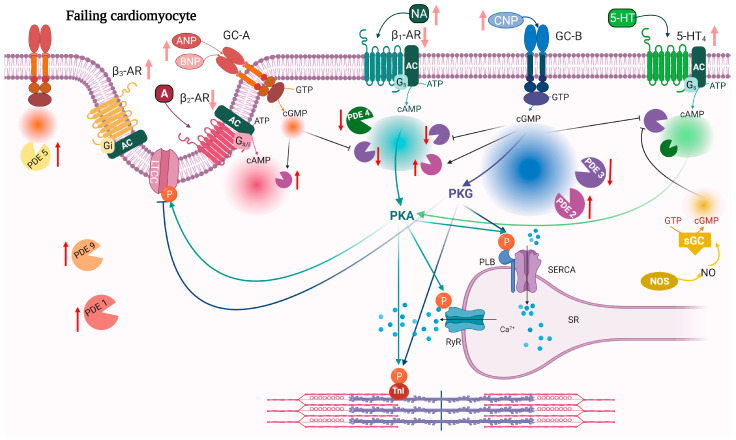
Remodeling and signaling alterations in heart failure. During HF, cardiomyocytes undergo structural remodeling which leads to alteration of the T-tubule structure, re-organization of the receptors and compartments. Red arrows indicate changes in expression of PDEs while light red arrows indicate changes in ligand and receptor expression in HF. 5-HT_4_ receptors are increased in HF and their stimulation induces a PIR and LR. Both GC-B and sGC stimulation increases cGMP that inhibits PDE3 activity towards 5-HT_4_-stimulated cAMP (light green). 5-HT_4_, 5-HT_4_ serotonin receptor; other abbreviations as in Figure 1. Created with BioRender.com.

**Table 1 ijms-23-02145-t001:** Modulation of PDEs in heart failure.

PDE	Expression	Activity	Localization	References
PDE2	↑	↑	↑ activity around β_2_-AR and PLB↓activity around β_1_-AR	[134,141,180]
PDE3	↓ ↑	↓ ↑	↑ activity around β_1_-AR	[141,181,182,183,184,185]
PDE4	↓	↓	↓PDE4D activity around RyR	[40,63,76,79,183,186]
PDE5	↑	↑	From Z-line localization to diffuse localization	[187,188,189,190,191]
PDE9	↑	↑		[105]
PDE10	↑	↑		[3]

Upregulated for an arrow that points up, and downregulated for the arrow that points down.

## Data Availability

Not applicable.

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
