# Peer review of "Phosphodiesterases and Compartmentation of cAMP and cGMP Signaling in Regulation of Cardiac Contractility in Normal and Failing Hearts"

_ijms, 2022, doi:10.3390/ijms23042145_

Round 1

Reviewer 1 Report

This is an excellent contribution to the role of cAMP and cGMP in the regulation of cardiac contractility and heart function. It describes the microcompartimentation of synthesis and degradation of the two types of cyclic nucleotides, the different activities and function of the respective degrading enzymes, the PDEs.

We can understand that a clinical application of these data is not so easy as we previously thought.

I cannot find any shortcomings and would suggest publication of the manuscript as it is.

Thanks for giving to me the opportunity to read this complex manuscript.

Author Response

This is an excellent contribution to the role of cAMP and cGMP in the regulation of cardiac contractility and heart function. It describes the microcompartimentation of synthesis and degradation of the two types of cyclic nucleotides, the different activities and function of the respective degrading enzymes, the PDEs.

We can understand that a clinical application of these data is not so easy as we previously thought.

I cannot find any shortcomings and would suggest publication of the manuscript as it is.

Thanks for giving to me the opportunity to read this complex manuscript.

Response to reviewer 1:

Thank you very much for reviewing our manuscript. We appreciate your comments.

Reviewer 2 Report

This work is well designed and presented,

I would like to propose some minor revisions:

  • Figures 1 and 2 present poorly legible writings.
  • Some acronyms described are no longer used, such as NCX, MCU, PMCA .., I propose to delete them; In addition, the acronyms ANP, BNP, and CNP are described in parentheses.
  • Line 189; the authors wrote: "Some studies" but there is only one reference.
    For a better approach to the relevant content of this work, I propose a summary table in which the modulations of the PDEs in Heart Failure are listed.

Author Response

This work is well designed and presented,

Thank you very much for reviewing our manuscript.

I would like to propose some minor revisions:

  • Figures 1 and 2 present poorly legible writings.

We have now increased the size of the text in Figure 1 and Figure 2.

  • Some acronyms described are no longer used, such as NCX, MCU, PMCA .., I propose to delete them; In addition, the acronyms ANP, BNP, and CNP are described in parentheses.

Thank you for pointing this out.

We have deleted the acronyms NCX, MCU and PMCA (line 89-90) and also revised the acronyms for ANP, BNP and CNP (line 117-119).

  • Line 189; the authors wrote: "Some studies" but there is only one reference.

Thank you for pointing this out. We have changed the text accordingly (now line 204-205)

  • For a better approach to the relevant content of this work, I propose a summary table in which the modulations of the PDEs in Heart Failure are listed.

We agree with the reviewer and have inserted a table that summarizes the changes in heart failure.